



# Seismological assessment of human activity levels

# during the COVID-19 pandemic

Jeongin Lee[1] and Tae-Kyung Hong[1*]

**Affiliation and address:**

[1]Yonsei University, Department of Earth System Sciences, 50 Yonsei-ro, Seodaemun-gu

Seoul 03722, South Korea.

**Correspondence to:**

tkhong@yonsei.ac.kr (Tae-Kyung Hong)





## Abstract

The COVID-19 virus has a high infection rate, spreading fast in the world. Lockdown and stay-at-home actions have been taken in many countries to reduce the rate of the virus spreading. The daytime ambient seismic noises in 11 major cities of 7 countries are assessed. Daytime seismic noises in 10 am to 6 pm at frequencies $\geq 2$ Hz are assessed. The seismic noise levels are compared with the community mobility data that represent the human activities. The high-frequency seismic noise levels present high correlation with the human activities. The human activities decrease with the number of daily confirmed cases. The peak noise-level reductions in lockdown periods were as high as 42-96 %. The noise levels generally started to decrease since the days when the daily confirmed cases reached ∼500. The noise level variation presents the lockdown progress. The noise level recovers with time since the end of lockdown. The high correlation between seismic noise level and community mobility suggests possible utilization of seismic noises for anonymous monitoring of human activities.



# 1 Introduction

COVID-19 spreads fast in the world. The high infectivity of the virus changes human life. The World Health Organization (WHO) recommended social distancing to slow down the spreading rate. Social distancing and wearing a mask may be the only ways to reduce the infection rate. Many countries locked down cities and shutdown workplaces to block the spread of the virus. However, the effectivity and effective maintenance period of social distancing is unknown. Further, it is difficult to assess the level of social activity of people.

Fluid population is crucial information to be collected. Realtime mobility data may be useful to identify the fluid population at certain locations. Location-based information is necessary to account the level of lockdowns. Big data from location-based service (LBS) for mobile internet can be applied in this regard. However, the information from location-based service may suffer from privacy infringement that is a critical issue in data acquisition. Further, publicly open big data is limited available.

Ambient seismic noise represents ground responses to a composite effect of microseismic sources that include anthropogenic (cultural) sources such as traffic, construction, cultural activity, and industrial operations (Groos and Ritter, 2009; Larose et al., 2015; Riahi and Gerstoft, 2015; Coward et al., 2003; Fuchs et al., 2018). Human activity is a major source of ambient seismic noises greater than 2 Hz (Plesinger and Wielandt, 1974; Kar and Mohanty, 2006; Bokelmann and Baisch, 2008; Hong et al., 2020). Human activities produce high-frequency ground vibrations that decay fast with distance (Aki and Wu, 1988; Hong et al., 2005; Diaz, 2016). High-frequency ambient seismic noise field develops by scattering in the crust (Aki and Wu, 1988; Sato and Fehler, 1997; Hong and Kennett, 2003, 2004; Hong et al., 2004).

High-frequency ambient seismic noises may represent the level of human activities (Groos and Ritter, 2009; Hong et al., 2020). Recently, it was reported that high-frequency seismic-noise levels are correlated with economic growth (Hong et al., 2020). The noise levels may be used for realtime monitoring of economic condition. Thus, seismic noises may be useful to treat the personal identity anonymously for assessment of human activities.

The lockdowns in COVID-19 pandemic provide a chance to examine the correlation between seismic noise level and human activity level (Lecocq et al., 2020; Denolle and


Nissen-Meyer, 2020; Dias et al., 2020). We examine the utility of seismic noises as a tool
to assess the human activities without privacy infringement.

## 2 Data and COVID-19

The first confirmed case of COVID-19 virus infection was reported in Wuhan, China on
December 31, 2019. We analyze the ambient seismic noise levels before and after the
virus outbreak and social distancing (Gibney, 2020). We consider 7 representative countries
including China, South Korea, France, Spain, Italy, the UK, and the USA. China and South
Korea had the first and second largest numbers of confirmed cases by March 9, 2020,
respectively. China and South Korea experienced high contagion followed by a recession
in the infection rate. The contagion cases were concentrated in local regions: Daegu and
Gyeongsangbuk-do Province in South Korea and Wuhan and Hubei Province in China. France,
Spain, Italy, the UK, and the USA experienced fast spreading after China and South Korea.
Further, the countries are distributed in Asia, Europe, and North America of the northern
hemisphere that may have comparable seasonal effects in virus spreading (Fig. 1).
Enshi in Hubei Province, China was locked down on January 24. Hubei Province was
released from lockdown on March 25. The first confirmed case in South Korea occurred on
January 20 (Fig. 1). The number of confirmed cases increased rapidly following the mass
contagion in Sincheonji Church, Daegu, on February 18. Daegu experienced a high increase in
confirmed cases after February 21. The confirmed cases in South Korea are concentrated in
Daegu and Gyeongsangbuk-do Province. Stay-at-home action was taken in Daegu on February
20. High-intensity social distancing setting has been in effect since March 23 in South Korea.
Many countries enacted lockdowns or equivalent governmental actions to reduce outdoor
activities. Italy, France, and the UK went into lockdown on March 10 (northern Italy on March
9), March 17, and March 23, respectively. Spain and New York in the USA declared a state of
national emergency on March 15 and March 20, respectively. We collect the information on the
global confirmed cases from from the World Health Organization (https://covid19.who.int/)
and Korea Centers for Disease Control & Prevention (http://ncov.mohw.go.kr/en/). The
numbers of confirmed cases generally decrease after lockdowns with time delays (Fig. 1).
China and South Korea present rapid decay in the numbers of daily confirmed cases. Italy,




France, and Spain present mild decreases in the numbers of daily confirmed cases, while USA
and UK display slow decays (Fig. 1).
We select 11 cities of the seven countries where high rates of virus infection were reported in
the early pandemic. The cities are Paris (France), Marseilles (France), Madrid (Spain), Milan
(Italy), Rome (Italy), Edinburgh (UK), New York (USA), Los Angeles (USA), Seoul (South
Korea), Enshi (Hubei Province, China), and Beijing (China). We collect continuous vertical
records of broadband or short-period seismic stations in the cities in 7 countries where high
infection rates have occurred (Fig. 1). The seismic records are collected from the Incorporated
Research Institutions for Seismology and the Korea Meteorological Administration. The
sampling rates are 40, 50 or 100 Hz. The analyzed periods are December 2019 to May 2020 for
the cities in China (Enshi and Beijing) and February 2020 to May 2020 for the other cities.
Community mobility data provide information of human activities in local regions. We
collect the human mobility data from Google (https://www.google.com/covid19/mobility/).
The Google mobility data is available since mid-February, 2020. The mobility data for grocery
and pharmacy business decrease mildly since the pandemic, recovering faster than other
business types. The mobility data for retail and recreation business, workplace, and transit
stations vary similarly one another.

## 102  3   Seismic noise analysis

We examine the spectra of continuous vertical velocity records (Fig. 2). Spectral composition
of seismic noises is site-dependent. Anthropogenic noises are primary components in ambient
seismic-noise field in frequencies $\geq 2$ Hz (Coward et al., 2003; Fuchs et al., 2018; Hong et al.,
2020). The vertical velocity spectra present weak ambient noises at frequencies $\geq \sim 2$ Hz since
the virus outbreak and social distancing (Fig. 2). The weakening of ambient noise is observed
in wide high frequency ranges. The noise weakening recovers since lockdown release.
Considering the spectral contents of seismic noises, we choose frequency bands of 5-15
Hz for analysis (Fig. 3). We analyze seismic records at frequencies 2-4 Hz for stations with
incidental high-frequency noises (stations BJT and ENH) (Fig. 3). Seismic noises from the
other stations are analyzed in frequencies of 5-15 Hz. We calculate the power spectral density





(PSD) based on 2-hour time windows that are shifted by 1 hour with 50 % overlap (Fig. 4). The seismic noise levels present daily periodicity and diurnal variations.

The diurnal variation in high-frequency noise levels resembles the diurnal cycle of human activities. Daytime noise levels are higher than nighttime noise levels (Groos and Ritter, 2009; Diaz, 2016) (Fig. 5). The noise levels on weekdays are stronger than those on weekends and holidays. The seismic noise levels decrease temporarily in lunch time.

We assess the daytime ambient seismic noises at 10 am to 6 pm to represent the daily noise levels (Fig. 6). The seismic noise levels are high in daytime and low in nighttime. We determine the representative noise levels by stacking the daytime PSDs (Fig. 7). The daily noise levels present weekly periodicity. We use only the weekday noise levels excluding the noise levels in weekends. The noise levels present apparent noise decrease during weekends (Fig. 8). An analysis of weekday noise level may enable us to assess the level of social and economic activities (Hong et al., 2020).

## 4   Ambient noise level variation

We observe fast decay in seismic noise levels after January 14. The ambient noise levels in Enshi and Beijing decreased rapidly before the city lockdown (Fig. 9). The low noise level in Enshi continued until mid-March and started to increase before the city release on March 25. On the other hand, the noise level in Beijing started to increase gradually in early February, and recovered fully in mid-April.

The seismic stations in Beijing and Enshi, China present low seismic noises in late January to March. The ambient seismic noises in frequencies greater than 2 Hz display similar feature (Fig. 9). The daytime noise levels in Seoul decreased mildly between January 30 and March 9, after which it gradually recovered (Fig. 9).

The noise levels in Rome and Milan, Italy, decreased after March 9. Similar features have been observed in Madrid since March 9, Edinburgh since March 16, New York and Los Angeles since March 12, and Paris and Marseilles since March 17. It is intriguing to note that the noise levels in most regions started to decrease even before the lockdown or equivalent governmental actions were enacted. This observation suggests that people might have concern about the





fast spreading of the virus in the regions, reducing their outdoor activities spontaneously. The
noise level decreased further after governmental actions.
The noise levels dropped by 42-96 % relative to the usual daily noise levels on weekdays in
the countries that experienced lockdown or equivalent governmental actions (Fig. 9). The noise
level in Seoul decreased only by 9 %, recovering gradually with decreasing daily confirmed
cases in South Korea.

## 5 Correlation with mass mobility data

We compare the daily noise level changes with the human mobility volume changes of various
business (Fig. 9). The human mobility decreases as telecommuting and shutdown of workplaces
increase. The seismic noise levels present correlations with the mobility data. The levels of
correlations between the ambient noises and mobility data are different by business type.
Also, the magnitude of noise-level decrease is different by region. This may be partly because
the medium responses to the human activities and composition of human activities and
populations are different by region.
The high correlation between noise level and human mobility suggests that the
high-frequency seismic noises are mainly excited by human activities (Groos and Ritter, 2009;
Larose et al., 2015; Hong et al., 2020; Gibney, 2020). The decreasing seismic noises suggest
decreasing ground-motion inducing sources. The seismic noise levels and human activities
decrease with the number of daily confirmed cases (Fig. 10). The correlation suggests that
both the noise level and mobility data may represent the human activities reasonably. The
seismic noises may be useful for monitoring of human activity, keeping anonymity.

## 6 Discussion and conclusions

The noise-level decrease suggests effective social distancing. The daily confirmed cases started
to decrease in 11-32 days after the effective social distancing (gray boxes in Fig. 10). This
observation suggests that social distancing may be an effective way to reduce the infection
rate. It is noteworthy that the daily confirmed cases increased continuously for some times
(i.e., 11-32 days) after the effective social distancing. The observation suggests that the social



distancing has the reserve time of two weeks to one month to be effective for reducing the
daily confirmed cases.
The number of cumulative confirmed cases appears to be correlated with the time lapsed
until the effective social distancing. However, the lapse time is different by country; the
noise-level decrease (i.e., effective social distancing) took place mostly around the first date
of 500 or more daily confirmed cases in the country (UK, USA, France, Spain, and Italy).
The noise-level decay rate may represent the level of public participation (lapse times in
red boxes in Fig. 10). A large decay rate of noise level may suggest high public participation.
The effectivity of social distancing may be dependent on the level of social distancing and
public participation. This observation suggests that confirmed cases may start to decrease
sooner if social distancing is enacted earlier.
The high-frequency seismic noise levels are reasonably represented by mobility data. The
relative influence of human activities on seismic noises is different by city. It is noteworthy
that human mobility data is limitedly available due to the privacy infringement. The high
correlation with mobility data suggests that the ambient noise may be used for realtime
monitoring for the human mobility without privacy infringement. This observation suggests
that the seismic noise data may replace the big data information.

## Data availability

The data and results of this study will be available on Dryad (https://datadryad.org/stash)
when the paper is published.

## Author contributions

JL collected data, performed analyses, and prepared figures. TKH led the research, guided
the analyses, developed the methods, and wrote the manuscript. JL and TKH discussed the
results.



## Competing interests

The authors declare that they have no conflict of interest.

## Acknowledgments

The seismic data were collected from the Incorporated Research Institutions for Seismology (IRIS, https://www.iris.edu) and the Korea Meteorological Administration (KMA, https://www.weather.go.kr).

## Financial support

This work was supported by the Korea Meteorological Administration Research and Development Program under grant KMI2018-02910. Additionally, this research was partly supported by National Research Foundation of Korea (NRF-2017R1A6A1A07015374, NRF-2018R1D1A1A09083446).

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



**Figure 1.** (a) Map of 11 seismic stations in 7 countries and temporal variations of confirmed cases for (b) China, (c) Italy, (d) France, (e) United States of America, (f) United Kingdom, (g) Spain, and (h) South Korea. Continuous vertical seismic records are collected from the seismic stations. The daily and cumulative numbers of confirmed cases are presented. The lockdown starting dates and released dates are marked.

**Figure 2.** Temporal variation of spectral amplitudes: (a) map of stations and periods, and vertical spectrograms for stations (b) BJT, (c) ARBF, and (d) RMP. The lockdown starting dates and released dates are indicated. The first dates of 10, 50, 100, 500 daily confirmed cases are marked. The seismic noises decrease apparently during the lockdown periods.

**Figure 3.** Spectral contents of ambient seismic noises before the COVID-19 outbreak at stations (a) RMP in Rome and (b) ENH in Enshi. Ambient seismic noises in frequencies ≥2 Hz present daily periodicity and diurnal variations associated with human activities. Frequency bands of 5-15 Hz or 2-4 Hz are used for seismic noise analysis.

**Figure 4.** Vertical power spectral density (PSD) variation at frequencies of 5-15 Hz or 2-4 Hz in stations (a) BJT in Beijing, (b) ENH in Enshi, (c) RMP in Rome, (d) MILN in Milan, (e) ARBF in Marseilles, (f) S1108 in Paris, (g) CPNY in New York, (h) USC in Los Angeles, (i) EDI in Edinburgh, (j) UCM in Madrid, and (k) SEO in Seoul. Power spectral densities of seismic noises are presented. The cumulative numbers of confirmed cases are presented. The noise levels are low in March and April in most stations.

**Figure 5.** Vertical power spectral densities in frequencies of 5-15 Hz at stations (a) RMP in Rome and (b) CPNY in New York in February 2020 before the virus outbreak in Italy. Weekends (Saturdays, Sundays) are marked. The seismic-noise amplitudes are large in weekdays, and small in weekends.

**Figure 6.** Diurnal variation of seismic noise amplitudes in weekdays at stations (a) RMP in Rome and (b) CPNY in New York. The analyzed daytime seismic noises are marked. The daytime noise levels are larger than the nighttime noise levels. The seismic noises are weak in lunchtime.



**Figure 7.** Daily average seismic noise levels at frequencies at frequencies of 5-15 Hz or 2-4 Hz in stations (a) BJT in Beijing, (b) ENH in Enshi, (c) RMP in Rome, (d) MILN in Milan, (e) ARBF in Marseilles, (f) S1108 in Paris, (g) CPNY in New York, (h) USC in Los Angeles, (i) EDI in Edinburgh, (j) UCM in Madrid, and (k) SEO in Seoul. Power spectral densities of seismic noises are presented. The cumulative numbers of confirmed cases are presented.

**Figure 8.** Representative daily seismic noise variation at stations (a) RMP in Rome and (b) CPNY in New York. The noise levels in weekends are excluded to avoid the weekend effect. The noise levels in weekends are presented for comparison.

**Figure 9.** Comparison between seismic noise level changes and mobility data at stations (a) RMP in Rome, (b) MILN in Milan, (c) ARBF in Marseilles, (d) S1108 in Paris, (e) CPNY in New York, (f) USC in Los Angeles, (g) EDI in Edinburgh, (h) UCM, Madrid, and (g) SEO in Seoul. The daily numbers of confirmed cases are presented.

**Figure 10.** Ambient noise-level changes and daily confirmed cases in 7 countries (UK, USA, France, Spain, Italy, South Korea, and China). The temporal variation of seismic noise levels (solid line) in 5-15 Hz (2-4 Hz for Beijing and Enshi) is compared with daily confirmed cases (histogram). The seismic noise level decreases after the COVID-19 outbreak. The noise level reduction ($\Delta L$) varies between -96 and -9 %. The number of daily confirmed cases reduces in 11-32 days after the noise-level decrease. The dates of lockdown or equivalent governmental actions (blue arrow) and lockdown release (red arrow) are marked. The first dates of 10, 50, 100 and 500 daily confirmed cases ($N_{10}$, $N_{50}$, $N_{100}$, and $N_{500}$) are annotated.
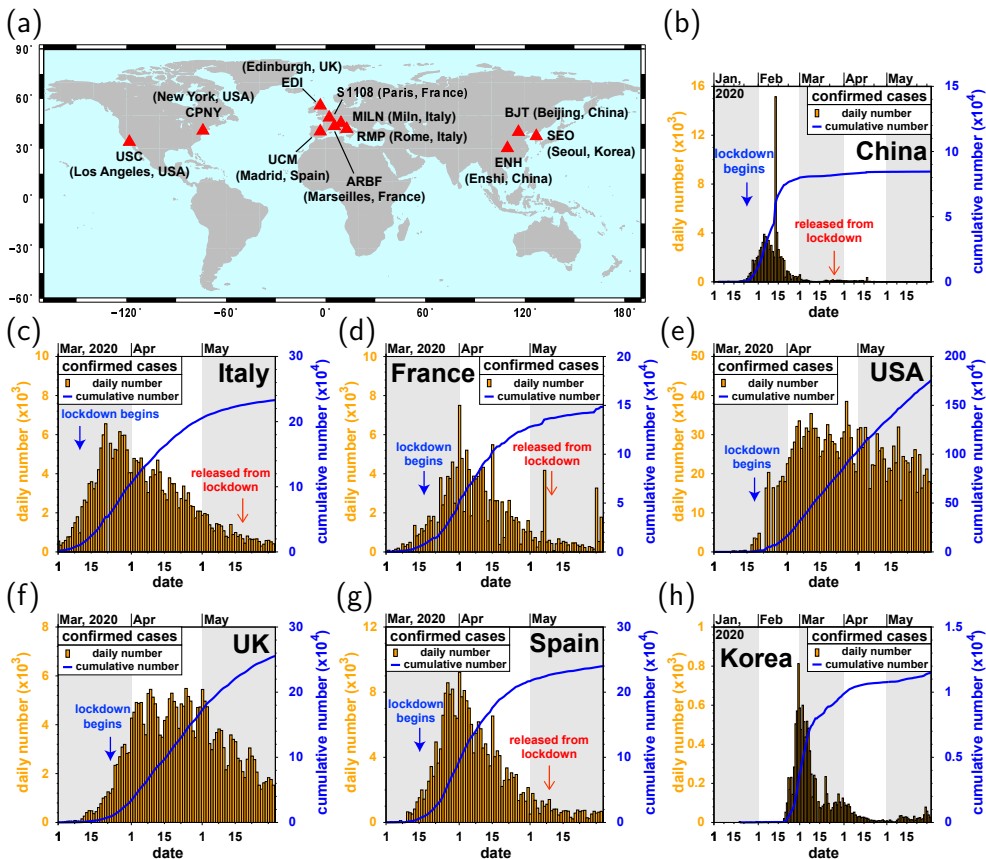

**Figure 1.** (a) Map of 11 seismic stations in 7 countries and temporal variations of confirmed cases for (b) China, (c) Italy, (d) France, (e) United States of America, (f) United Kingdom, (g) Spain, and (h) South Korea. Continuous vertical seismic records are collected from the seismic stations. The daily and cumulative numbers of confirmed cases are presented. The lockdown starting dates and released dates are marked.



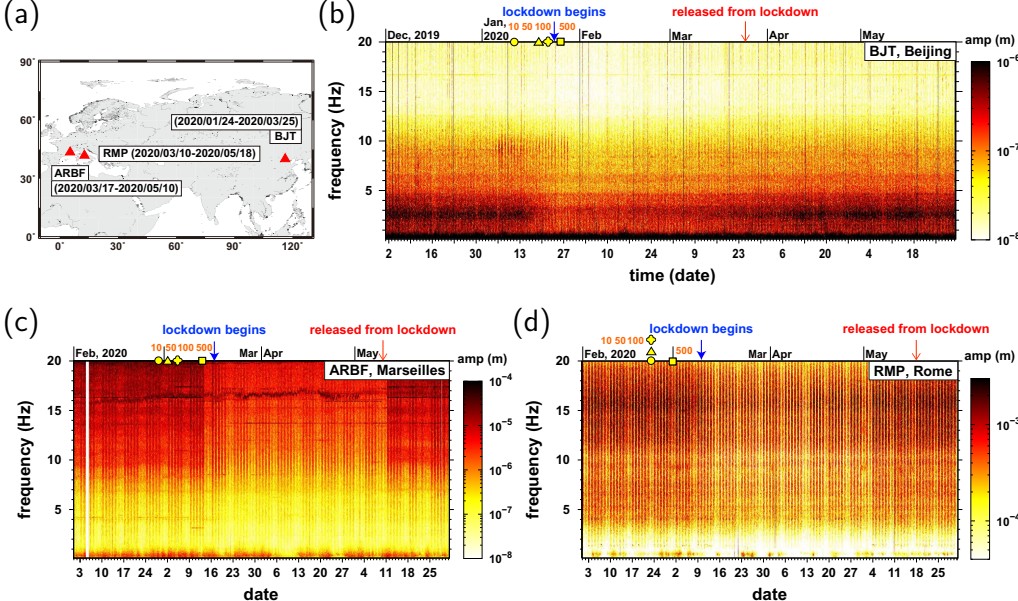

**Figure 2.** Temporal variation of spectral amplitudes: (a) map of stations and periods, and vertical spectrograms for stations (b) BJT, (c) ARBF, and (d) RMP. The lockdown starting dates and released dates are indicated. The first dates of 10, 50, 100, 500 daily confirmed cases are marked. The seismic noises decrease apparently during the lockdown periods.


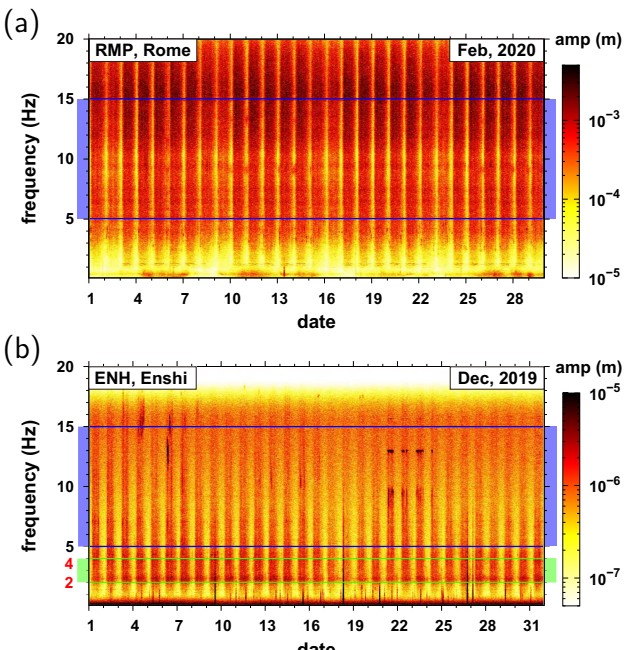

**Fig. 3.** Spectral contents of ambient seismic noises before the COVID-19 outbreak at stations (a) RMP in Rome and (b) ENH in Enshi. Ambient seismic noises in frequencies ≥2 Hz present daily periodicity and diurnal variations associated with human activities. Frequency bands of 5-15 Hz or 2-4 Hz are used for seismic noise analysis.



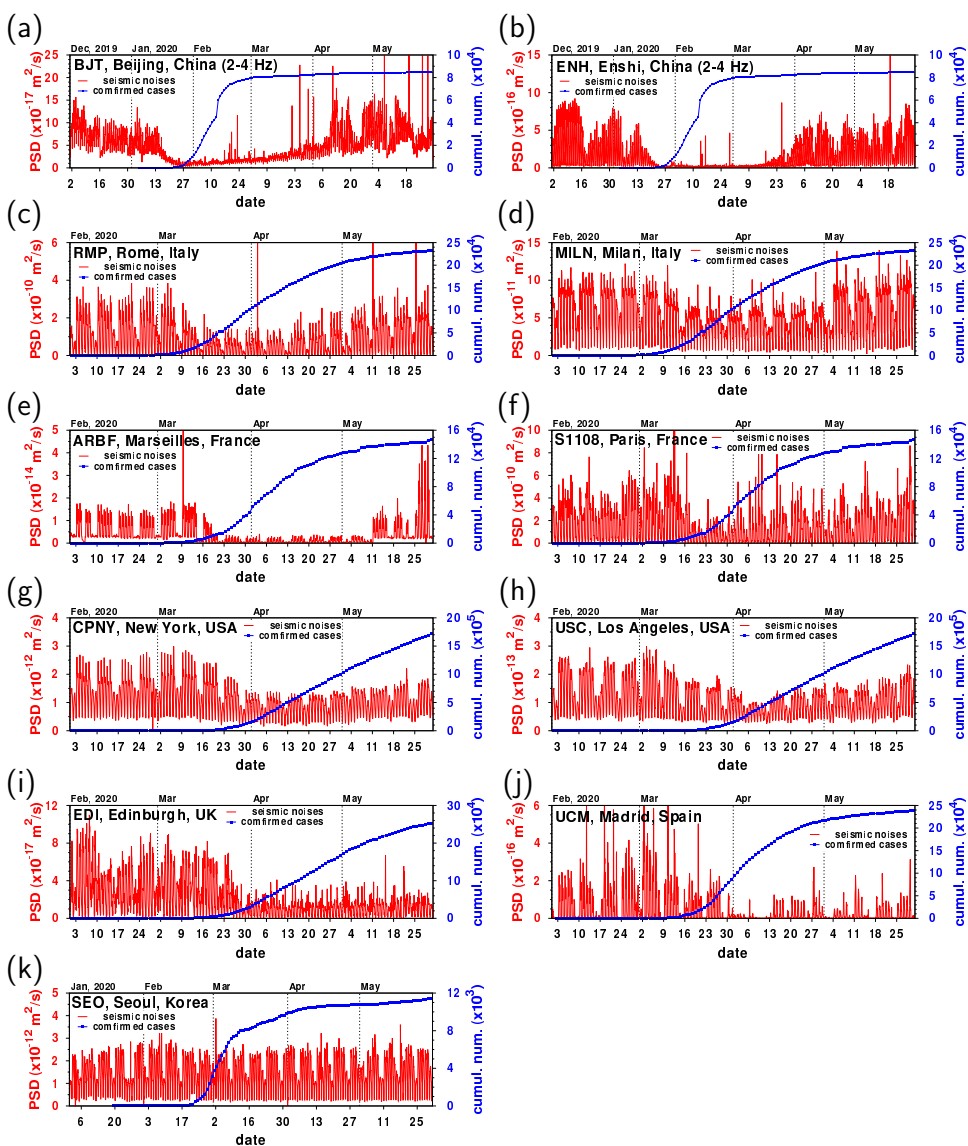

**Figure 4.** Vertical power spectral density (PSD) variation at frequencies of 5-15 Hz or 2-4 Hz in stations (a) BJT in Beijing, (b) ENH in Enshi, (c) RMP in Rome, (d) MILN in Milan, (e) ARBF in Marseilles, (f) S1108 in Paris, (g) CPNY in New York, (h) USC in Los Angeles, (i) EDI in Edinburgh, (j) UCM in Madrid, and (k) SEO in Seoul. Power spectral densities of seismic noises are presented. The cumulative numbers of confirmed cases are presented. The noise levels are low in March and April in most stations.





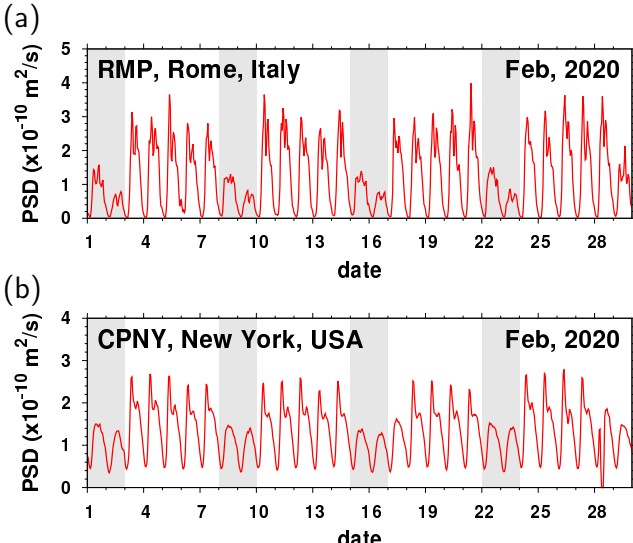

**Fig. 5.** Vertical power spectral densities in frequencies of 5-15 Hz at stations (a) RMP in Rome and (b) CPNY in New York in February 2020 before the virus outbreak in Italy. Weekends (Saturdays, Sundays) are marked. The seismic-noise amplitudes are large in weekdays, and small in weekends.





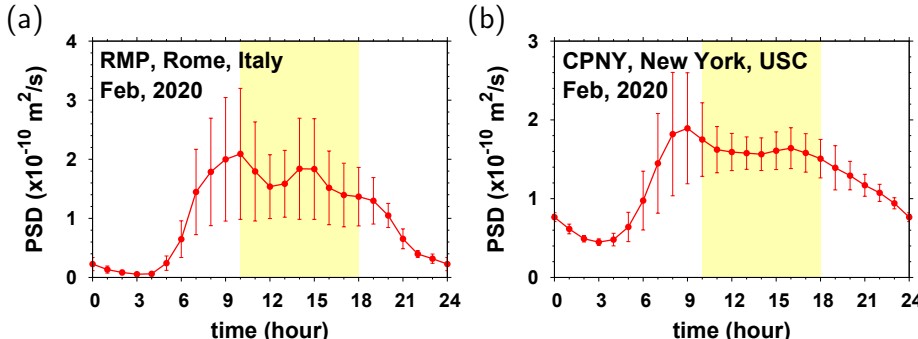

**Fig. 6.** Diurnal variation of seismic noise amplitudes in weekdays at stations (a) RMP in Rome and (b) CPNY in New York. The analyzed daytime seismic noises are marked. The daytime noise levels are larger than the nighttime noise levels. The seismic noises are weak in lunchtime.





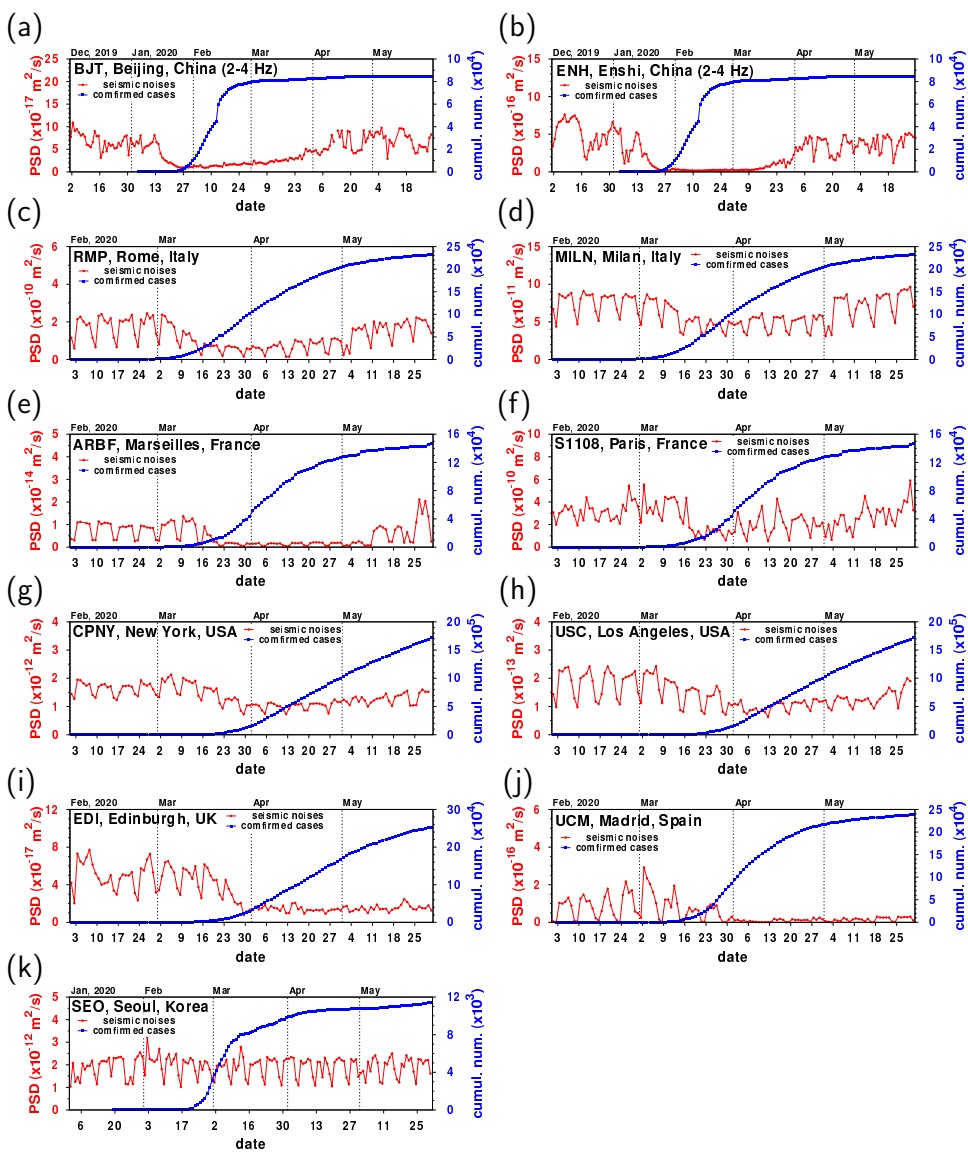

**Figure 7.** Daily average seismic noise levels at frequencies at frequencies of 5-15 Hz or 2-4 Hz in stations (a) BJT in Beijing, (b) ENH in Enshi, (c) RMP in Rome, (d) MILN in Milan, (e) ARBF in Marseilles, (f) S1108 in Paris, (g) CPNY in New York, (h) USC in Los Angeles, (i) EDI in Edinburgh, (j) UCM in Madrid, and (k) SEO in Seoul. Power spectral densities of seismic noises are presented. The cumulative numbers of confirmed cases are presented.





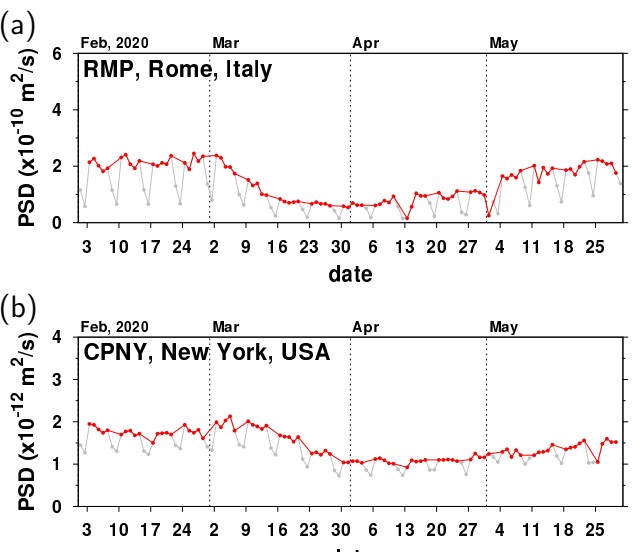

**Fig. 8.** Representative daily seismic noise variation at stations (a) RMP in Rome and (b) CPNY in New York. The noise levels in weekends are excluded to avoid the weekend effect. The noise levels in weekends are presented for comparison.



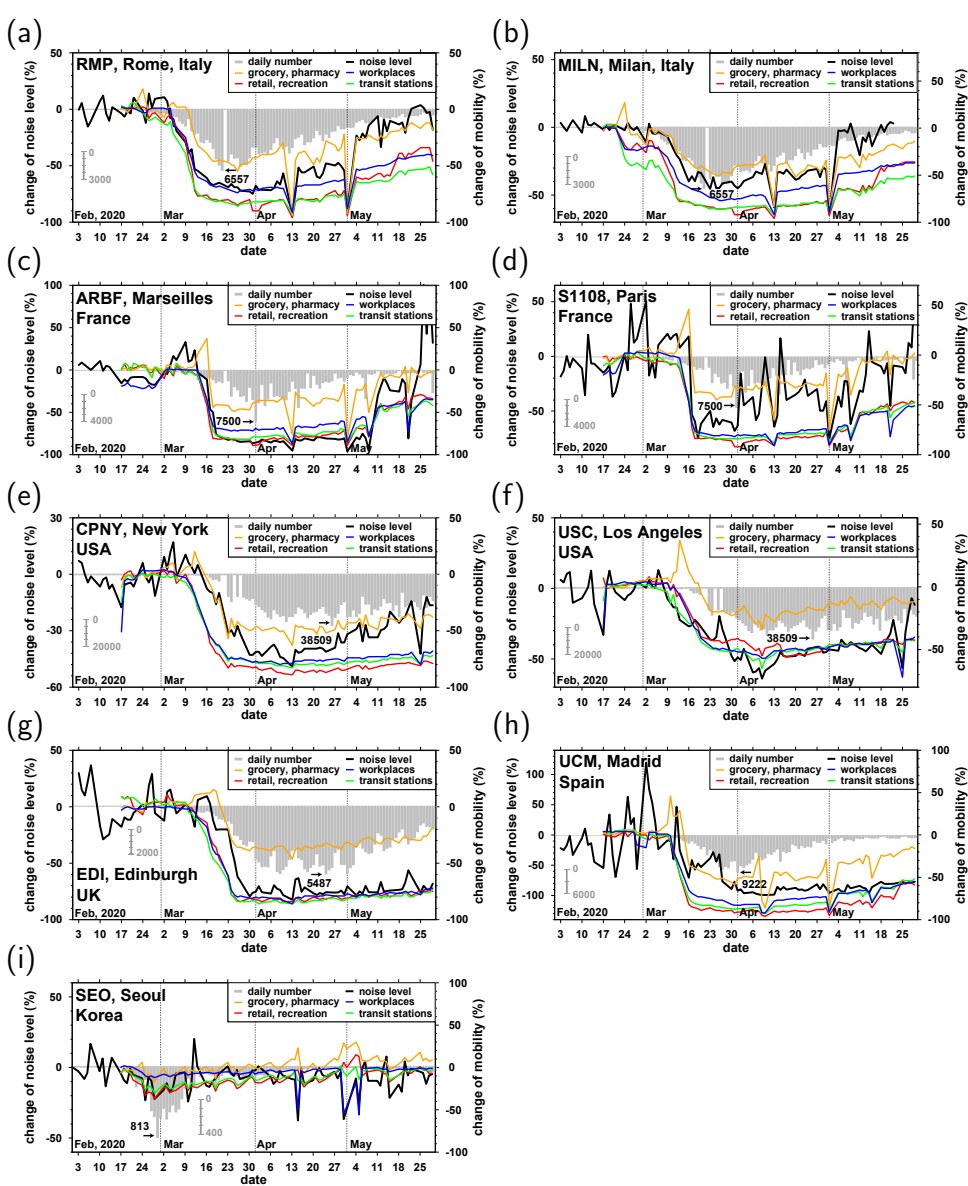

**Figure 9.** Comparison between seismic noise level changes and mobility data at stations (a) RMP in Rome, (b) MILN in Milan, (c) ARBF in Marseilles, (d) S1108 in Paris, (e) CPNY in New York, (f) USC in Los Angeles, (g) EDI in Edinburgh, (h) UCM, Madrid, and (g) SEO in Seoul. The daily numbers of confirmed cases are presented.


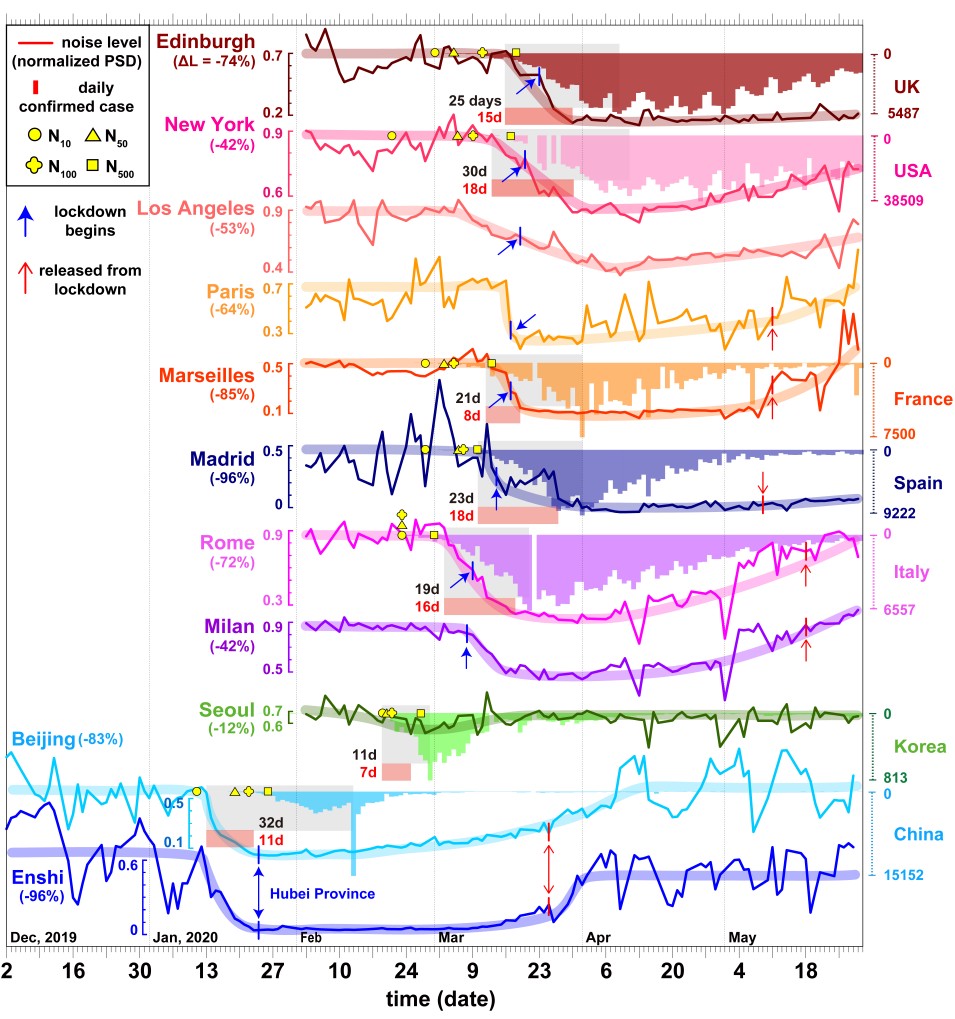

**Figure 10.** Ambient noise-level changes and daily confirmed cases in 7 countries (UK, USA, France, Spain, Italy, South Korea, and China). The temporal variation of seismic noise levels (solid line) in 5-15 Hz (2-4 Hz for Beijing and Enshi) is compared with daily confirmed cases (histogram). The seismic noise level decreases after the COVID-19 outbreak. The noise level reduction ($\Delta L$) varies between -96 and -9 %. The number of daily confirmed cases reduces in 11-32 days after the noise-level decrease. The dates of lockdown or equivalent governmental actions (blue arrow) and lockdown release (red arrow) are marked. The first dates of 10, 50, 100 and 500 daily confirmed cases ($N_{10}$, $N_{50}$, $N_{100}$, and $N_{500}$) are annotated.