# Peer review of "Seismological assessment of human activity levels"

_Solid Earth, 2020_

## Referee Comment (RC1) · Anonymous Referee #1 · 14 Dec 2020

This manuscript presents an analysis of the seismic noise levels at 11 stations located in cities around the world, to discuss the relationship with the quieting periods following COVID-19 pandemic lockdown measures. The text is very short (less than 6 pages in my PDF) and divided in 6 sections, some of them including just a couple of paragraphs. The authors review some general statement on the pandemic, report the seismic stations used without providing details, describe the seismic noise analysis (again, without details), its time variation and the correlation with mobility data. The final discussion section focuses more on the evolution of the cumulative number of COVID19 cases than in seismic noise. On the other hand, the number of figures is high, some of them clearly reiterative or unnecessary.

[Figure]

The relationship between mobility and seismic noise has already been documented in a number of papers, including a reference one published in Science (Lecocq et al, 2020). Of course, there is room for more contributions on the subject, but in my opinion, these new contributions should provide additional information, not merely report the existence of a relationship that we already know.

My main concern with this manuscript is that 6 of the 11 presented stations have already been analyzed in previous contributions, showing the same kind of spectrograms, PSDs and mean value graphs. Stations BTJ, ENS, RMP and MILN are discussed in detail in Xiao et al, SRL, 2020; SEO has been analyzed by Hong et al, SRL, 2020 and spectrograms for CPNY are included in Lecocq et al, Science, 2020 (by the way, MILN data has also been presented in Poli et al., Sci Rep., 2020 and Piccini et al, Sci Rep., 2020). Additionally, 2 more of the sites (USC and EDI) have been included in Lecocq et al 2020, although without presenting the details of the analysis. Therefore, to my knowledge, only 3 of the sites provide unpublished results; Paris, Marseille and Madrid. Note that the Xiao et al, SRL, 2020 paper, where 4 of the stations discussed also here is not included as a Reference.

Regarding the previously published results, I've not detected any new analysis or conclusion compared to what have been described previously.

Regarding the new sites, the main problem is that neither of them is located in the cities that they are supposed to represent. ARBF is in the countryside, at more than 20 km from Marseille; UCM is located also in the countryside, at more than 40 km from Madrid city center. Finally, S11018 is a RaspberryShake instrument located at the outskirts of Rambouillet, a relatively small city located at 40 km of Paris. We are not taking of being located more or less close to the city cente, but from large distances that completely bias any relationship between seismic data and ground motion in the city. A usual reader will trust the authors and will not check if the stations are really in the city or not. I found that referring to these sites as "Paris", Marseille" and "Madrid" is absolutely unacceptable.

A second important point has to do with the large number of inaccuracies and errors all along the manuscript, as the use of "PSD" to refer to mean values of something (in the figures the labels are in m2/s; may be accelerations in m/s2 ??). No information is provided on the kind of instrumentation used in each case. No references to the parametrization for the calculation of PSD nor to the software used to do so. No tables with the location and characteristics of the used stations, a section is entitled "correlation.." but only include a subjective comparison, some statements are included without justification (pe. Lines 96-101, some sentences repeated in the same paragraph etc...

Finally, the discussion is very poor as does not include any conclusion derived from the analyzed data which could be of utility for the audience.

As a conclusion, in my opinion, this manuscript has to be rejected, as, although it addresses relevant scientific questions, it does not present new concepts and does not provide new relevant conclusions. The lack of information can result in a misinterpretation of some of the results /stations located at 20-40 km from the reference city. Relevant references are missing and the language is far from being fluent and precise

I don't see any possibility of reaching the minimum quality and novelty required to publish a paper in a journal that wants to be a worldwide reference.
* * *

---

## Referee Comment (RC2) · Anonymous Referee #2 · 19 Dec 2020

The manuscript looks at the variation of high-frequency seismic noise levels at 11 seismic stations in 7 countries following the implementation of COVID-19 lockdown measures. Those observations are compared with the respective Google community mobility data and daily confirmed infection cases for the corresponding country.

The quality of English is severely lacking throughout the manuscript to the point that it occasionally becomes hard to understand (e.g. "Fluid population" - line 33, "Thus, seismic noises may be useful to treat the personal identity anonymously for assessment of human activities." - line 52).

The scope of the manuscript is very general and not sufficiently defined. Although the

title indicates an interest in studying human activity with seismic noise, the manuscript loosely addresses lockdowns, social distancing, confirmed infection cases, and ambient noise without comprehensively laying out the relationship between them.

The issue is illustrated at line 167: "The observation suggests that the social distancing has the reserve time of two weeks to one month to be effective for reducing the daily confirmed cases." It is not clearly explained (and unlikely) how such information comes out of observations of seismic noise variations. Such statements could indicate an intention to analyze the effectiveness of lockdown measures on the number of reported cases, which is not accomplished in this manuscript.

More problematic, the content of the manuscript echoes previous publications such as Lecocq et al. 2020 and Xiao et al. 2020 without offering further substantial contribution. For example, Chinese stations in Line 124 - 135 were already analyzed with more perspective on the impact of the Chinese new-year on the noise for the previous years.

From a more general standpoint, no background is given on the seismic stations; what type of site are they installed in, is it an urban or rural context, who operates them, etc. This latter point is also problematic as the data providers/networks are not credited at all although they should be even if the data is openly accessible.

As it stands, the manuscript is not suitable for publication and very unlikely to reach satisfying standards with significant corrections. It should therefore be rejected. I recommend the authors to substantially revisit the research from the beginning and to consider a specific scope, making sure that a potential future new manuscript would contain original content with a significant contribution. I also encourage them to have any future submissions carefully reviewed by a native English speaker before submission.

—————————————————